# OpenReview forum: "UniJEPA: Enhancing Robot Policy via Unified Continuous and Discrete Representation Learning"
_ICML.cc/2026/Conference — ICML 2026 regular_

### Official Review · Reviewer_dTxy · 2026-03-07

**Soundness:** 3
**Presentation:** 2
**Significance:** 3
**Originality:** 2
**Overall Recommendation:** 4
**Confidence:** 4

**Summary:**

This paper proposes a novel Vision-Language-Action (VLA) framework called UniCod for learning robotic policies from video data annotated with vision, language, and action signals.
* Problem: Previous VLA models in robotic tasks either suffer from degradation of foundational VLM(Vision-Language Model) capabilities when fine-tuned on limited robotic data, or fail to preserve vision-language alignment when leveraging generative models, making it difficult to generalize to novel scenes and task instructions.
* Solution: To address the limitations of previous methods, UniCod introduces a two-stage training process: (1) pretraining a unified vision-language embedding model that aligns continuous visual predictions with discrete language representation, and (2): fine-tuning this for action modeling in continuous action spaces.
* Methodological contribution: UniCod uses a Mixture-of-experts Transformer (MoT) architecture with a VLM backbone and incorporates techniques like bidirectional attention and future state prediction to preserve vision-language alignment while enabling effective policy learning.
* Empirical Results: Experiments are conducted on simulation benchmarks and real-world platforms, showing improvements over baselines in unseen tasks and strong generalization to novel objects and scenes.

**Compliance With Llm Reviewing Policy:**

Affirmed.

**Final Justification:**

While the initial submission suffered from minor typos and missing experimental results, the authors addressed all of these issues clearly in the rebuttal. I am therefore raising my score.

**Key Questions For Authors:**

1. Baseline comparison fairness
> Does the unification in Calvin benchmark to a single third-person input truly increase "fairness", or does it remove t he inherent advantages of baselines that originally use multi-view or history?

2. Language generalization extent
> In Appendix A.4. the paper only provides demos for unseen objects without any explanation of OOD language description. Could you clarify what evidence or results support the mention of OOD language description on page 7, Line 361?

3. Key equations explanation
> Could you provide a more detailed breakdown and intuitive explanation of the core equations in low matching ($ \mathcal{L}_{flow} $)?

**Limitations:**

The paper would benefit from a more discussion of limitations, particularly:

1. The paper's definition of "unseen" or OOD scenarios is relatively narrow compared to its broader claims. It primarily tests generalization to novel objects and color/background changes, but does not address more critical OOD challenges in VLA settings, including novel task composition, variations in linguistic phrasing (e.g., synonyms), and temporal delays.

2. The experiments are on a narrow set of tasks/manipulators; no tests on locomotion, outdoor robotics, reducing broader applicability.

**Strengths And Weaknesses:**

1. Strength
* Soundness
> * The methodology is logically structured with a clear two-stage approach, supported by mathematical formulations (e.g. losses for generative and flow-based branches, action prediction objectives).
> * Experiments are comprehensive, covering both simulation and real-world settings, including seen/unseen splits for robustness
> * Ablation study systematically test components like w/wo continuous and discrete features, pretraining paradigms, and vision encoders providing evidence for design choices.

* Presentation
> * The paper is well-organized with clear sections, figures, and tables.
> * Visualizations like t-SNE plots effectively illustrate feature evolution, and equations are neatly formatted.
> * Language is concise, with good flow from problem motivation to experiments.

* Significance
> * UniCod addresses a timely challenge in robotics: bridging internet-scale VLMs with embodied action learning, potentially enabling more scalable, generalizable policies.
> * Improvements in unseen tasks highlight its potential for real-world deployment.
> * By fusing continuous and discrete representations, it could influence broader AI fields like multi-modal learning. The open-sourcing promise adds practical value.

* Originality
> * The unified continuous-discrete pretraining is novel, combining VLM alignment with future prediction in a MoT framework using MoE for action heads.
> * Innovations like bidirectional masking in attention and interpolating actions in continuous space differentiate it from priors. Integrating VQA with robot data for pretraining is a fresh data strategy.

2. Weakness
* Soundness
> * Real-world evaluations rely on limited trials (e.g., 20-50 per task), which may not capture variability in noisy environments.
> * While the authors unified the baselines to a single third-person view for fair comparison, this choice may remove the original strengths of those baselines. Re-implementing prior works under a single-view setting makes it difficult to fully disentangle whether the reported gains stem from the proposed method's advantage or from placing the baselines in a less favorable setting.

* Presentation
> * Some sections feel rushed. For example, in section 3.2, description of flow matching loss lacks smoothness and could benefit from a more detailed explanation for non-experts.
> * In page 3, Line 148, it would be better to modify "optimaze" to "optimize"
> * In page 16, Line 843-848, it would be better to remove the template sentences.
> * It would be better fill in the running title.

* Significance
> * While significant for VLA research, the impact of the paper may be niche to robotics; it doesn't fundamentally shift paradigms in general AI. (e.g., no new theoretical insights beyond empirical tweaks.)

* Originality
> * Many elements build incrementally on existing work (e.g. PaliGemma backbone, MoE from recent VLMs, future prediction from VPP and World models).

---

> ### Author Rebuttal · Authors · 2026-03-29
>
> We sincerely thank the reviewer for their constructive feedback, detailed reading, and helpful suggestions. We address your specific concerns below.
>
> **1. Baseline Comparison Fairness and Single-View Setting (Soundness & Key Q1)**
> Our primary motivation for evaluating on the Calvin single-view (ABC-D) split is that generalizing from limited visual information (a single uncalibrated camera) is one of the most challenging and realistic open-world robotics scenarios. Furthermore, the baselines we compared against (GR-1, VPP, UP-VLA) do not incorporate specific architectural designs for multi-view processing. Thus, this setting provides a rigorous testbed demonstrating that UniCoD extracts more actionable spatial/dynamic information from the exact same limited input.
>
> To further address your concern, we have supplemented multi-view Calvin ABC-D results below. UniCoD consistently demonstrates a performance advantage, aligning with our real-world multi-view results (Sec 4.3).
>
> | Multiview Setting | 1 | 2 | 3 | 4 | 5 | Avg |
> |---|---|---|---|---|---|---|
> | GR-1 | 0.854 | 0.712 | 0.596 | 0.497 | 0.401 | 3.060 |
> | UP-VLA* | 0.948 | 0.890 | 0.839 | 0.792 | 0.715 | 4.184 |
> | VPP | 0.965 | 0.909 | 0.866 | 0.820 | 0.769 | 4.329 |
> | UniCoD | 0.993 | 0.955 | 0.893 | 0.842 | 0.788 | 4.471 |
> *(UP-VLA is our multi-view reproduction; VPP and GR-1 are cited from original papers).*
>
> **2. OOD Language Generalization (Key Q2 & Limitation 1)**
> We apologize for the omission in the appendix. During real-world evaluations, UniCoD successfully executed tasks with out-of-distribution (OOD) phrasing. For example, trained on "put the apple into the pink plate," it succeeded when prompted with "pick up the red fruit and place it into the transparent plate"—despite the transparent plate being completely unseen. We will add a dedicated Appendix subsection with these examples.
>
> To clarify, "unseen" strictly pertains to scene-level variations (novel backgrounds, objects, distractors) and diverse linguistic phrasing.
>
> Regarding the "temporal delay" concern, we measured inference speeds on an H20 GPU (average of 100 forward passes). UniCoD maintains a relatively low temporal delay suitable for real-time execution:
>
> | Approach | Latency (ms) |
> |---|---|
> | UniCoD (Ours) | 260 |
> | $\pi_0$ | 250 |
>
> **3. Intuitive Explanation of Flow Matching Loss (Presentation & Key Q3)**
> We agree Section 3.2 needs simplification and will expand it.
> *Intuitive Explanation:* Instead of predicting future actions in one step, Flow Matching learns a continuous transformation (a vector field) that gradually morphs a simple distribution (Gaussian noise) into the target distribution (predicted actions). $\mathcal{L}_{flow}$ trains the network to predict the "velocity" (or direction) needed to push the noise toward the true future representation. By minimizing the difference between the predicted vector field and the optimal straight-line path, the model learns a smooth, stable dynamics model in the latent space.
>
> **4. Scope, Originality, and Real-World Trials (Soundness, Significance, Originality & Limitations)**
> * **Originality:** While UniCoD builds upon existing components (like MoE and VLM backbones), our core contribution is the systematic and novel integration of these elements into a unified VLA framework. Specifically, using a Mixture-of-Transformers to independently handle understanding, continuous latent dynamics, and action prediction—without requiring pixel-level reconstruction—is a novel paradigm that significantly advances robotic representation learning.
> * **Limited Trials:** We agree that mitigating environmental noise in real-world evaluations is a critical factor. In fact, we accounted for this during our real-world testing phase. We allocated the number of evaluation trials (ranging from 20 to 50) based on the inherent randomness and complexity of each specific task. During tests, we strictly controlled environmental setups to ensure that the degree of randomness (e.g., object initial poses, distractors) was consistent across all policies, thereby achieving a fair possible comparison. We believe that 20-50 trials per task provide a statistically meaningful sample size that mitigates variance to a reasonable extent in physical robot experiments. We have plotted error bars in Figures 4 and 5.
> * **Narrow Scope:** While our evaluation focuses on manipulation, this is standard practice in VLA research due to massive differences in action spaces and data pipelines across domains. UniCoD targets generalist robotic manipulation, but we agree extending to locomotion/outdoor navigation is a critical next step and will discuss this in the Limitations.
>
> **5. Presentation and Typos**
> Thank you for your meticulous reading. We will:
> 1. Correct "optimaze" to "optimize" (Line 148).
> 2. Remove template sentences on page 16.
> 3. Fill in the running title.
> 4. Thoroughly polish Section 3.2 for readability.
>
> Thank you again for your time and effort in reviewing our work!

---

> > ### Author Rebuttal · Reviewer_dTxy · 2026-04-03
> >
> > We thank the authors for their detailed and thoughtful rebuttal, as well as for providing additional experimental results and clarifications. The responses were helpful and addressed most of our concerns.
> >
> > We particularly appreciate the new multi-view results on the Calvin benchmark, which strengthen the authors’ claims regarding performance advantage. The clarification on OOD language generalization, along with the promise to add concrete real-world examples in the appendix, adequately resolves our earlier ambiguity on this point. The intuitive explanation of the flow matching loss is also welcome and will improve accessibility.
> >
> > The authors’ defense regarding the number of real-world trials and environmental control is reasonable for physical robot experiments. We also acknowledge their point about the current standard practice in VLA research focusing primarily on manipulation tasks.
> >
> > After considering the rebuttal, we believe the authors have successfully addressed the major weaknesses we pointed out, especially concerning baseline fairness, language generalization evidence, and presentation issues. The additional multi-view results and planned revisions further improve the paper.
> >
> > While some limitations remain (particularly the relatively narrow task scope), the rebuttal has sufficiently mitigated our primary concerns.
> >
> > We thank the authors again for their diligent response.

---

> > > ### Author Response · Authors · 2026-04-04
> > >
> > > Dear Reviewer dTxy:
> > >
> > > We sincerely appreciate the time and effort you have taken to review our rebuttal. Could we politely ask if there are any further concerns existed? We are always willing to address any of your further questions. If there are no additional concerns, we sincerely wish you could reconsider your score.
> > >
> > > Thank you once again for your valuable time！
> > >
> > > Best Regards,
> > >
> > > The Authors

---

### Official Review · Reviewer_EYnf · 2026-03-10

**Soundness:** 4
**Presentation:** 3
**Significance:** 3
**Originality:** 2
**Overall Recommendation:** 5
**Confidence:** 3

**Summary:**

UniCoD proposes a unified understanding-generation-execution paradigm, which is built on the MoT architecture equipped with three modality-specialized expert modules for understanding, generation and action, and designs a targeted two-stage training process.

The first stage conducts large-scale pre-training on over 1 million web-scale instructional manipulation videos from robot and human demonstrations, with the core objectives of learning discrete language representations to achieve semantic understanding of embodied scenes and task instructions, and simultaneously acquiring continuous visual representations to accomplish robotic world modeling and future state prediction. The second stage performs fine-tuning on the VLA multimodal data collected from robot embodiments, whose core objectives are to train the action expert module, jointly optimize the prediction of continuous visual future states and the generation of robotic actions. The robotic policy is both strong generalization ability and efficient action execution capability, which significantly improves its performance in simulation environments and real-world out-of-distribution tasks.

**Compliance With Llm Reviewing Policy:**

Affirmed.

**Final Justification:**

After rebuttal, my concerns have been resolved,so I maintain my positive assesment.

**Key Questions For Authors:**

1. Compare the inference overheads.

The paper states in Section 3.3 that "Although the additional experts introduce extra parameters, they only increase the pre-training overhead and do not lead to higher inference latency", yet no explicit computational complexity metrics or inference speed (FPS) are provided to validate this claim.

2. Verify the effective retention of VLM’s understanding and generation capabilities.

Whether the core understanding and generation capabilities of the pre-trained VLM are effectively preserved after the completion of the full two-stage training process remains unaddressed and unvalidated with experimental evidence.

3. Evaluate the impact of severe parameter imbalance across modules.

In UniCoD, the VLM (2.3B parameters) accounts for nearly 80% of the total model parameters (2.9B), while the generation expert and action expert each only have 0.3B parameters, resulting in a severe imbalance in parameter allocation. There are two questions: (1) Whether this extreme parameter imbalance is likely to induce task execution errors in robotic manipulation? (2) Are there a large number of experimental cases where the model achieves correct semantic understanding of task instructions but produces deviated or incorrect robotic manipulation actions?

**Limitations:**

yes

**Strengths And Weaknesses:**

## strengths
1. This paper addresses a highly meaningful research problem: how to integrate action prediction with the preservation of the understanding capabilities inherent to VLMs.

2. The proposed method is validated by relatively comprehensive experiments, which are conducted on two simulation environments and two real-world robotic setups to ensure the generalizability of the results.


## weaknesses
1. The meaning of the parameter $h$ in $o_{t+h}$ is not explicitly defined in Section 3.1, which causes some confusion for readers. It is inferred that $h$ may denote the future $h$ time steps; thus, why does the language instruction $l$ also vary with time (as referenced by $l_{t+h}^{pred}$ in the Training Objective section)? I think it's a small typo.

2. The model architecture of UniCoD lacks notable novelty, as it bears strong similarities to Motus. Additionally, its core proposition an unified framework for continuous and discrete representation learning has already been explored in prior VLM research such as Bagel and Emu, making the core idea less innovative.

---

> ### Author Rebuttal · Authors · 2026-03-29
>
> We sincerely thank the reviewer for their detailed review and insightful questions. We address your concerns point-by-point below.
> 1. Clarification on Notation and Typo (Weakness 1)
> We apologize for the confusion caused by the notation in Section 3.1. You are absolutely correct that $h$ denotes the future time steps (also the horizon).
> Regarding $l_{t+h}^{pred}$, this is indeed a typo in the manuscript. The high-level language instruction $l$ provided by the user remains constant throughout the task execution and does not vary with time. We will correct $l_{t+h}^{pred}$ to simply $l$ (or denote it as the generated sub-task plan if referring to the understanding expert's output at step $t$) in the revised Training Objective section to ensure mathematical rigor.
>
> 2. Novelty Compared to Motus, Bagel, and Emu (Weakness 2)
> We acknowledge that unified understanding and generation have been explored in foundational VLM research like Bagel and Emu. However, their primary focus is on general-purpose image and text generation.
> In contrast, UniCoD does not aim to train a unified understanding-generation foundation model. Instead, our core innovation lies in utilizing world modeling in a continuous visual space to bridge the inherent gap between high-level VLM representations and low-level physical action spaces. The continuous visual predictions serve as an intermediate bridge, allowing UniCoD to effectively ground the VLM's semantic understanding into environmental dynamics and precise robotic control.
> Furthermore, compared to Motus (which also uses MoT):
> *   Motus uses the VLM merely as an encoder (not as a MoT expert), lacking explicit language reasoning and sub-task planning capabilities. UniCoD treats understanding/planning as an independent expert.
> *   Motus learns dynamics via computationally heavy pixel-level video reconstruction. UniCoD demonstrates that learning dynamics in a continuous latent space is far more efficient and sufficient for robust robotic policies (as analyzed in Appendix A.1 and ablation study).
>
> 3. Inference Overheads and FPS (Key Question 1)
> The inference latency of our model on an H20 GPU (average of 100 forward passes) is shown below. We will add a dedicated FPS table in the revision to explicitly validate efficiency.
> | Approach | Latency (ms) |
> | --- | --- |
> | UniCoD (Ours) | 260 |
> | pi0​ | 250 |
>
> 4. Retention of VLM's Capabilities (Key Question 2)
> We appreciate this rigorous question. We evaluated the Visual Question Answering (VQA) capabilities---representing the understanding aspect---of the original PaliGemma (our VLM backbone), Pi0, Pi05, and the VLM within UniCoD across three typical benchmarks using the VLMEval toolkit. The results are presented in the table below. Notably, UniCoD, Pi0, and Pi05 all utilize PaliGemma as their VLM backbone, and the results demonstrate that UniCoD preserves the original pre-trained capabilities to the greatest extent.
>
> |             | **MME** | **MMMU** | **MMBench** |
> | ----------- | ------- | -------- | ----------- |
> | Paligemma   | 1670    | 34.7     | 65.3        |
> | pi0     | 0       | 2.1      | 1.7         |
> | pi05 | 979     | 15.7     | 29.4        |
> | UniCoD      | 1544    | 30.9     | 62.4        |
>
> Regarding generative capabilities, because our training relies on latent space visual prediction, it is not feasible to directly assess the generation quality through image reconstruction. Instead, we conducted ablation studies on this specific component within our experiments. The results prove that this generative capability significantly enhances performance during action execution, particularly by endowing the model with superior semantic generalization abilities (e.g., robustly handling novel, unseen target objects).
>
> 5. Impact of Parameter Imbalance (Key Question 3)
> The parameter allocation (2.3B frozen VLM vs. 0.3B experts) is an intentional design choice for robotic learning:
> - Why 0.3B Experts? Inference latency is critical for reactive behaviors. Large experts decrease control frequency. Additionally, since action datasets are relatively small, massive action experts risk overfitting, severely degrading zero-shot generalization.
> - Execution Errors: We do observe cases where the model perfectly understands instructions and moves correctly, but dexterous hand actions lack precision, causing grasp failures. This shows UniCoD has strong generalization but exhibits inaccuracies in fine-grained control, accounting for most failures. We will add a detailed discussion of this in the Appendix.
>
> Thank you again for your time and effort in reviewing our work!

---

> > ### Author Rebuttal · Reviewer_EYnf · 2026-04-01
> >
> > Thank you for the thoughtful rebuttal. I am satisfied that my main concerns have been addressed.

---

> > > ### Author Response · Authors · 2026-04-03
> > >
> > > We sincerely appreciate the time and effort you have taken to review our rebuttal. In response to your insightful feedback, we have carefully incorporated the requested comparison to other methods and included a discussion of the details of our model. We are always willing to address any of your further questions.

---

### Official Review · Reviewer_wDHF · 2026-03-10

**Soundness:** 3
**Presentation:** 3
**Significance:** 2
**Originality:** 3
**Overall Recommendation:** 5
**Confidence:** 4

**Summary:**

The paper studies generalist robot policy learning in settings where both semantic understanding and future-state modeling are important for manipulation. It proposes UniCoD, a vision-language-action framework that jointly learns discrete language/understanding representations and continuous future visual representations within a mixture-of-transformers architecture, using large-scale pretraining on robot and human instructional videos followed by embodiment-specific action fine-tuning. The evaluation covers two simulation benchmarks (Calvin and SimplerEnv) and two real-world platforms (a Franka Panda arm and a 12-DoF dexterous hand), with comparisons to recent VLA and prediction-based baselines. The reported results indicate improvements over these baselines in simulation and on real-world seen and unseen tasks, including stronger performance on out-of-distribution object generalization.

**Compliance With Llm Reviewing Policy:**

Affirmed.

**Final Justification:**

The authors have successfully addressed my concerns. I will raise my rating from Weak accept to Accept.

**Key Questions For Authors:**

1. Refer to weakness 2, can UniCoD extend to multi-view setting to increase the applicable scope of the method?

**Limitations:**

yes

**Strengths And Weaknesses:**

**Strength**

1. The paper introduces a well-motivated unified formulation that combines discrete embodied understanding/planning supervision with continuous future visual feature prediction, rather than relying solely on tokenized generation or standard VLM-to-action fine-tuning. This is implemented concretely through a two-stage training pipeline, a modality-specialized mixture-of-transformers design, and distinct objectives for language prediction, future-feature regression, and flow-matching-based action modeling.
2. The experimental evaluation is notably broad, spanning both simulation and real-world settings across multiple embodiments, which provides substantive evidence about the method’s applicability. Specifically, UniCoD is evaluated on Calvin, two SimplerEnv robot settings, a 7-DoF Franka arm, and a 12-DoF dexterous hand, and the reported numbers show consistent gains.
3. The ablation studies directly test the paper’s central design claims and provide useful evidence for both pretraining and continuous visual prediction. For instance, Table 4 reports that removing pretraining reduces the average success on the real-world XArm setting from 76.7 to 60.8, while Table 6 shows that adding continuous prediction and then large-scale pretraining yields progressive gains on SimplerEnv-WindowX from 49.8 (without continuous prediction) to 60.3 (raw-pixel prediction) to 69.3 and 71.0 for the UniCoD variants.

**Weakness**

1. The title “Unified Continuous and Discrete Representation” attracted me to review this paper. However, it actually refers to the unification of textual and visual modalities, rather than a unified representation of discrete and continuous action spaces (e.g., discrete as in OpenVLA and continuous as in RoboFlamingo).
2. This paper built under single third-view setting as shown in Table 1. However, at present, robotic arms rely on multi-view information to perform fine operation tasks.

---

> ### Author Rebuttal · Authors · 2026-03-29
>
> We sincerely thank the reviewer for their time and for finding our paper interesting. We address your concerns regarding the terminology and the multi-view capabilities of our framework below.
>
>
> ---
>
> **1. Clarification on *Continuous and Discrete Representation* (Weakness 1)**
>
> We appreciate the reviewer pointing out this potential source of confusion. We completely understand that in the broader robotic learning literature, the terms "continuous" and "discrete" are frequently associated with action spaces (e.g., discrete action tokens in OpenVLA vs. continuous actions in RoboFlamingo).
> In the context of our paper, however, these terms describe the multimodal representation space:
> *   Discrete: Refers to the discrete textual/language tokens used by the understanding expert for high-level semantic reasoning and subtask planning.
> *   Continuous: Refers to the continuous latent visual features (extracted via ViT) used for spatial understanding and future dynamics prediction (as analyzed in Appendix A.1).
>
> *(Note: Our actual action output space is continuous).*
> We agree that the title and terminology might mislead readers expecting a discussion on action spaces. To prevent any misunderstanding, we will explicitly define what we mean by "continuous and discrete representations" early in the Introduction and Abstract of the revised manuscript.
>
>
> ---
>
> **2. Multi-View Capability and Extension (Weakness 2 & Key Question 1)**
>
> We apologize for the confusion caused by Table 1. Table 1 primarily focuses on a specific simulation benchmark setting that is designed to evaluate single-view generalization. Here, we have supplemented the experimental results for all models on the multi-view Calvin ABC-D tasks. As shown, UniCoD consistently demonstrates a performance advantage, which aligns with its superior performance observed in the real-world multi-view tasks in Sec 4.3. (Results for UP-VLA are based on our multi-view reproduction, while those for VPP and GR-1 are cited from their original publications.)
>
> | Multivew setting | 1     | 2     | 3     | 4     | 5     | Avg   |
> | ---------------- | ----- | ----- | ----- | ----- | ----- | ----- |
> | GR-1             | 0.854 | 0.712 | 0.596 | 0.497 | 0.401 | 3.060 |
> | UP-VLA*           | 0.948 | 0.890 | 0.839 | 0.792 | 0.715 | 4.184 |
> | VPP              | 0.965 | 0.909 | 0.866 | 0.820 | 0.769 | 4.329 |
> | UniCoD           | 0.993 | 0.955 | 0.893 | 0.842 | 0.788 | 4.471 |
>
> Importantly, UniCoD is not restricted to a single third-view setting and already supports multi-view inputs.
> As detailed in our manuscript in Appendix A.3 (XArm Dexterous Manipulation), our real-world dexterous manipulation experiments are actually conducted using a **dual-view visual input** (combining both first-person/egocentric and third-person cameras) to successfully perform fine, complex dexterous manipulation tasks.
>
> Architecturally, extending our framework to multiple views is straightforward and native to our design. Images from multiple cameras are processed through the frozen Vision-Language Encoder to extract their respective continuous visual features. These feature sequences are then spatially concatenated (often with camera-specific positional embeddings) before being fed into the Transformer Policy. This allows the policy to naturally integrate and attend to multi-view spatial information for precise control.
>
>
> ---
>
> Thank you again for your time and effort in reviewing our work! We hope our clarification can solve all your concerns, and we are always ready to answer any further questions!

---

> > ### Author Rebuttal · Reviewer_wDHF · 2026-04-03
> >
> > The authors have successfully addressed the concerns raised in my review.
> > In addition to the above, I wonder what the specific differences are between the continuous features predicted by the intermediate gen expert in UniCoD and standard VAE features.
> > Thanks for the thoughtful rebuttal to my additional question, I will raise my rating.

---

> > > ### Author Response · Authors · 2026-04-03
> > >
> > > Thank you for this insightful question. In fact, using a VAE to reconstruct future images is the approach adopted by most previous predictive VLAs (e.g., UP-VLA, InternVLA-A1). Regarding feature analysis, we have provided an intuitive visualization and discussion in Appendix A.1, where we observe that ViT features can more distinctly differentiate changes in action patterns within the same episode. Additionally, we have supplemented an experiment on the Calvin setting where we replaced the continuous visual prediction features in UniCoD with a VAE encoder. The comparison is as follows:
> > > |               | 1     | 2     | 3     | 4     | 5     | Avg   |
> > > | ------------- | ----- | ----- | ----- | ----- | ----- | ----- |
> > > | $pi_0$        | 0.915 | 0.778 | 0.650 | 0.557 | 0.460 | 3.360 |
> > > | $pi_{0.5}$    | 0.941 | 0.845 | 0.743 | 0.650 | 0.544 | 3.723 |
> > > | VAE Feature | 0.895 | 0.829 | 0.780 | 0.735 | 0.664 | 3.903 |
> > > | UniCoD        | 0.973 | 0.895 | 0.823 | 0.752 | 0.670 | 4.110 |

---

### Official Review · Reviewer_VbR6 · 2026-03-11

**Soundness:** 3
**Presentation:** 3
**Significance:** 2
**Originality:** 2
**Overall Recommendation:** 3
**Confidence:** 4

**Summary:**

This work proposes a unified vision–language–action (VLA) framework, termed UniCoD, which adopts a two-stage training strategy: first pretraining robotic reasoning and representation, followed by fine-tuning on manipulation demonstration datasets. This design enhances policy learning by jointly leveraging discrete task comprehension and continuous prediction of future robotic states. UniCoD is evaluated in both simulation and real-world environments, followed by ablation studies to analyze the individual contributions of each component.

**Compliance With Llm Reviewing Policy:**

Affirmed.

**Final Justification:**

The rebuttal effort is appreciated, sincerely; however, the explanation remains insufficient to fully support the claimed representation learning perspective.

Although the paper states that it approaches the problem strictly from a representation learning standpoint, the current presentation suggests that the learned representations primarily serve as a means to improve the action policy, rather than being investigated as a central object of study. This creates a mismatch between the stated motivation and the actual contribution.

Moreover, key discussions regarding representation learning are placed in the appendix rather than the main paper, which weakens the visibility and clarity of this contribution.

Last but not least, the analysis of representation learning is limited to empirical performance gains (e.g., contribution analysis), while deeper insights are lacking, such as a theoretically grounded understanding or verifiable properties of the learned representations.

The reviewer hope above mentioned comments could help to improve the quality of this work.

**Key Questions For Authors:**

1. Could the authors clarify the critical distinctions between UniCoD and other unified or MoT-based (UP-VLA, Motus, MoTVLA, InternVLA-A1...) approaches?

2. Could the authors clarify the motivation of employing $pi$0 instead of $\pi$0.5 series , as including the more recent $\pi$0.5 variants would likely provide a stronger and fairer comparison ($\pi$0.5 and $\pi$0.5_KI also output the subtask planning).

**Limitations:**

It would be beneficial for the authors to discuss the failure modes of the proposed approach, as well as the failure recovery behaviors.

**Strengths And Weaknesses:**

**Strengths**:
1. The study involves extensive pretraining and fine-tuning datasets, and the substantial experimental effort is acknowledged.
2. Diverse domains (simulation and real-world) and tasks (Pick&Place, Stack, and so on) are employed to evaluate the feasibility and effectiveness of the proposed UniCoD framework.
3. The manuscript is clearly presented and well written.

**Weakness**:
1. The primary concern lies in the novelty of the work. It seems that the main difference between this work and an existing work named "UP-VLA" is the architecture, which they transfer from one foundation model to mixture-of-transformers (MoT), while the main claim remains almost similar: "enhanced high-level semantic comprehension and low-level spatial understanding leads to better manipulation performance". Moreover, at this point, several MoT-based VLA works already exist, which reduces the perceived novelty of proposed UniCoD.

2. As a MoT-based VLA framework, the manuscript lacks a comprehensive review of prior work on MoT-based and unified VLA approaches. A more thorough discussion of related methods is necessary to position UniCoD within the existing literature properly. In addition, the selection of baselines in the experimental evaluation is not fully convincing. The current comparisons lack several relevant VLA methods that share similar design principles or architectural characteristics, such as $\pi$0.5 (and its variant $\pi$0.5_KI) and InternVLA-A1.

3. There is a concern that pretraining the understanding expert using predefined templates, such as the prefixes “Next subtask is:” and “Planning:”, may limit the generalization capability of UniCoD, particularly with respect to general and open-ended reasoning. Reliance on fixed prompt structures could potentially bias the model toward specific reasoning patterns, thereby reducing its flexibility in more diverse or open-ended scenarios.

---

> ### Author Rebuttal · Authors · 2026-03-29
>
> We sincerely thank the reviewer for their constructive feedback, recognizing our extensive pretraining/fine-tuning efforts, diverse evaluation domains, and clear presentation. We address your concerns below.
> 1. Clarification of Novelty and Contributions (Weakness 1)
> We agree the description of our novelty could be more precise. Our core contributions are:
> - Novel Training Paradigm & Efficient Dynamics Learning: We introduce a joint training paradigm for VLA tasks, simultaneously training embodied language understanding and dynamics prediction. Unlike prior works relying on costly pixel-level future generation, our framework learns future predictions directly in the latent space via continuous visual representations. This preserves high-level semantics efficiently, offering a novel VLA pre-training approach that circumvents the need for large-scale, action-annotated datasets.
> - Tailored MoT Architecture: While Mixture-of-Transformers (MoT) is not new, we uniquely use it to independently handle understanding (planning), dynamics modeling, and action prediction. This design perfectly suits our joint training paradigm, preserving pretrained multimodal capabilities while enabling task-specific adaptation without interference.
> - Real-World Generalization: Benefiting from continuous representation learning, UniCoD significantly enhances generalization. Real-world experiments (Figs 4 & 5) show substantial improvements when handling novel objects or operating with distractors, proving its superiority in unstructured environments.
> 2. Distinctions from MoT-based and Unified VLAs (Weakness 1, 2 & Key Q 1, 2)
> We will add a "Related Work" subsection to comprehensively review these approaches.
> - vs. UP-VLA & InternVLA-A1: UP-VLA uses a single LLM for autoregressive generation, while InternVLA-A1 uses MoT. UniCoD adopts a fundamentally different paradigm, using continuous visual features for visual predictions with open-source and self-annotated data. This continuous feature alignment yields superior performance (see Appendix A.1 and table below).
> - vs. Motus: Motus uses a VLM only as an encoder, lacking explicit language reasoning. It also learns dynamics via video reconstruction. UniCoD proves robust visual representations can be learned in a continuous latent space without pixel reconstruction.
> - vs. pi0,pi05,& MoTVLA: In addition to utilizing language data, UniCoD introduces a large-scale future feature prediction task. The improvements of UniCoD over pi0 in both simulation and real-world experiments strongly validate the superiority of this approach. To provide a stronger and fairer comparison (Q2), we have reproduced InternVLA-A1 and pi05 under the settings of Table 1. The updated results clearly demonstrate the effectiveness and competitive edge of our method against these recent baselines. (InternVLA-A1 is our re-implementation using a VAE encoder-decoder. Pi series​ uses official PyTorch implementations. All trained for 22k steps.)
> | Model | 1 | 2 | 3 | 4 | 5 | Avg |
> | --- | --- | --- | --- | --- | --- | --- |
> | pi0​ | 0.915 | 0.778 | 0.650 | 0.557 | 0.460 | 3.360 |
> | pi05​ | 0.941 | 0.845 | 0.743 | 0.650 | 0.544 | 3.723 |
> | InternVLA-A1* | 0.895 | 0.829 | 0.780 | 0.735 | 0.664 | 3.903 |
> | UniCoD | 0.973 | 0.895 | 0.823 | 0.752 | 0.670 | 4.110 |
>
> 3. Baseline Selection and pi05​ (Weakness 2 & Key Q 2)
> We initially chose pi0​ as a direct baseline because it allows an apples-to-apples comparison under identical training settings, isolating the effectiveness of our continuous feature prediction and discrete language planning. We excluded pi05​ initially as it was pre-trained on a proprietary dataset. However, to strengthen our results, we have now included pi05​'s performance in simulation (table above), further validating our approach.
> 4. Generalization with Predefined Templates (Weakness 3)
> To mitigate formatting biases from customized QA templates, we incorporated diverse template formats (e.g., planning, new subtask) during data construction. Additionally, we co-trained with VQA data (LLaVA-Pretrain) to preserve the VLM's pretraining capabilities. Consistent output patterns allow the model to focus on learning actual semantic content during embodied fine-tuning without disrupting inherent reasoning.
> 5. Limitations and Failure Modes
> We will add a Failure Analysis and Limitations section in the appendix.
> - Failure Modes: Primary failures occur during grasping (e.g., smooth, spherical toys requiring continuous angle adjustments). Conversely, the model shows excellent semantic generalization with novel/distractor objects, rarely moving incorrectly.
> - Failure Recovery: UniCoD exhibits partial recovery in closed-loop execution; it can replan and re-attempt grasps if the object remains in the workspace. However, it lacks an active visual search mechanism if the object moves out of view. We will note this as future work.
> We hope these clarifications address your concerns and will incorporate them into the final version.

---

> > ### Author Rebuttal · Reviewer_VbR6 · 2026-04-03
> >
> > The reviewer thanks the authors for providing the rebuttal explanation.
> >
> > While the authors attempt to distinguish their work from prior MoT-based approaches, the use of joint training and visual dynamic learning is not particularly novel at this stage. Although the authors emphasize training in the latent space, such techniques are already widely adopted in MoT-based methods. Furthermore, Motus does not rely on explicit language reasoning, as it also focuses on aligning generation and action within a latent space.
> >
> > The reviewer expected the motivation to be grounded in a representation learning perspective, as suggested by the title, “Unified Continuous and Discrete Representation Learning.” However, for a contribution positioned in representation learning, the manuscript should provide deeper insights and analysis of the learned representations, rather than primarily focusing on application-oriented metrics such as success rate.
> >
> > Nevertheless, the reviewer acknowledges the authors’ effort in conducting extensive real-world experiments and considers the concerns regarding novelty to be more a matter of personal research preference rather than a critical issue with the work. The reviewer will maintain the original score, as the theme of this conference is more aligned with machine learning. However, the reviewer believes the evaluation might be more favorable if the work were submitted to a robotics-focused conference.

---

> > > ### Author Response · Authors · 2026-04-03
> > >
> > > Dear Reviewer VbR6,
> > >
> > > We appreciate your prompt response and your acknowledgment of our extensive real-world experiments. We also appreciate your candor in stating that your remaining concerns regarding novelty stem from a "personal research preference." We would like to clarify the following critical points:
> > > 1. Distinct Novelty in Latent Space Training and Explicit Language Reasoning (vs. Motus)
> > > You mentioned that latent space training and joint training are not novel. We must clarify how our latent space training fundamentally differs from existing works.
> > > - Lightweight Feature Prediction vs. Heavy Reconstruction: In UniCoD, the latent space training specifically refers to our visual prediction (world modeling) mechanism. While Motus relies on computationally heavy pixel-level video reconstruction, UniCoD introduces a lightweight, JEPA-style continuous feature prediction scheme. This is a distinct and highly efficient paradigm for learning dynamics.
> > > - Explicit Language Capabilities: As you correctly noted, Motus uses a VLM merely as an encoder. In contrast, UniCoD utilizes the VLM as an independent expert within our architecture. This is not a trivial difference; it ensures UniCoD retains explicit language reasoning capabilities. To empirically prove this, we evaluated UniCoD on standard VQA benchmarks. As shown in the table below, UniCoD successfully preserves strong language abilities.
> > >
> > > |             | **MME** | **MMMU** | **MMBench** |
> > > | ----------- | ------- | -------- | ----------- |
> > > | Paligemma   | 1670    | 34.7     | 65.3        |
> > > | $\pi_0$     | 0       | 2.1      | 1.7         |
> > > | $\pi_{0.5}$ | 979     | 15.7     | 29.4        |
> > > | UniCoD      | 1544    | 30.9     | 62.4        |
> > >
> > > 2. Representation Learning of Robotics MUST be Grounded in Task Execution
> > >
> > > We agree that a paper focused on representation learning should provide deeper insights into the representations. However, our title reflects our core motivation: addressing the bottlenecks of action execution in current VLA paradigms from a feature representation perspective.
> > > - First, we did provide insights into the representations. In Appendix A1, we compared different feature types and demonstrated that continuous visual features provide superior guidance for action models. Also, our extensive experimental results show that our insight are functional in the robotics manipulation area.
> > > - Second, and most importantly, our experiments explicitly reveal a critical insights that the **joint prediction of language and continuous visual features significantly empowers the VLA model with strong semantic generalization capabilities in Out-of-Distribution (OOD) scenarios**. This is a direct testament to the quality of the learned representations. In Embodied AI, the ultimate proof of a learned representation (especially rich semantic VL features) must be grounded in physical execution. High success rates in OOD settings are not merely "application metrics"; they are the most rigorous and objective proof that the underlying representation learning is effective, aligned, and truly generalizable. Evaluating embodied representations in a vacuum without testing their impact on actual task capabilities would be fundamentally incomplete.
> > >
> > > 3. Conference Fit: Embodied and Robotics Learning is a Core ML Topic in ICML conference.
> > >
> > > The core challenges in modern VLAs and robotics fundamentally intersect with machine learning capabilities. We approach the problem strictly from a representation learning perspective, proposing a novel joint training paradigm and architecture, and we validate its effectiveness through robotic manipulation. Developing new ML architectures (like our unified continuous/discrete MoT) and proving their efficacy in embodied agents is a highly relevant, mainstream, and rapidly growing topic within the ICML community.
> > >
> > > Given that we have addressed the technical concerns, provided new baselines proving SOTA performance, and clarified that the remaining doubts are based on "personal research preference," we respectfully urge the reviewer to re-evaluate our work based on its objective technical merits and empirical strength.

---

### Decision · Program_Chairs · 2026-04-30

**Decision:**

Accept (regular)

**Comment:**

The reviewers liked the proposed UniCoD framework for its well-motivated unified formulation, which effectively bridges high-level semantic planning with low-level continuous dynamics prediction. They noted the extensive experimental effort, spanning diverse simulation environments and real-world robotic setups as a significant strength that provides substantive evidence of the method's applicability and generalizability. However, some reviewers were concerned about the incremental technical novelty compared to existing mixture-of-transformers (MoT) work like Motus and UP-VLA, the potential for overfitting due to predefined prompt templates, and the fairness of single-view baseline comparisons.

The rebuttal addressed the majority of the reviewers’ points by providing additional experimental results on multi-view benchmarks, demonstrating superior retention of VLM capabilities compared to current state-of-the-art models (e.g., Pi0), and clarifying the distinct efficiency of latent-space feature prediction over heavy pixel-level reconstruction.

After the rebuttal, the paper received two accept scores, one weak accept score, and one weak reject score. In the subsequent discussion phases, Reviewers wDHF, EYnf, and dTxy found the rebuttal satisfactory, particularly appreciating the authors' additional multi-view results and the quantitative evidence of semantic generalization. However, Reviewer VbR6 maintained reservations regarding the novelty of the work from a representation learning perspective, pointing out that the analysis primarily focused on application-oriented success rates rather than theoretical insights into the learned latent space. Conversely, Reviewer wDHF and EYnf supported the paper's acceptance, highlighting the significance of grounding VLM reasoning in physical execution and the robustness of the two-stage training paradigm.

ACs have read the paper, reviews, rebuttal, and discussion. ACs found the arguments in favor of the paper’s empirical significance and system integration to be persuasive, weighting the paper's demonstrated performance and extensive real-world validation over the concerns regarding theoretical depth. While the components build upon existing paradigms, the paper demonstrates a systematic and effective integration that enables state-of-the-art performance in complex, unstructured environments. Notably, the preservation of pre-trained VLM reasoning alongside fine-grained control is a critical challenge in VLA research, and UniCoD provides an interesting and efficient solution.

ACs reached a consensus on acceptance. The authors are encouraged to improve the final paper version by following reviewer recommendations, especially by moving the representation analysis and failure mode discussions from the appendix to the main text, clearly defining the “continuous and discrete” terminology to avoid confusion with action spaces, and ensuring the technical distinctions from prior MoT-based VLAs are explicitly stated.